# Best practices for spatial language data harmonization, sharing and map creation—A case study of Uralic

Timo Rantanen[1]*, Harri Tolvanen[1], Meeli Roose[1], Jussi Ylikoski[2], Outi Vesakoski[3,4]

1 Department of Geography and Geology, University of Turku, Turku, Finland, 2 Giellagas Institute for Saami Studies, University of Oulu, Oulu, Finland, 3 Department of Biology, University of Turku, Turku, Finland, 4 Department of Finnish language and Finno-Ugric Linguistics, University of Turku, Turku, Finland

* timo.rantanen@utu.fi

## Abstract

Despite remarkable progress in digital linguistics, extensive databases of geographical language distributions are missing. This hampers both studies on language spatiality and public outreach of language diversity. We present best practices for creating and sharing digital spatial language data by collecting and harmonizing Uralic language distributions as case study. Language distribution studies have utilized various methodologies, and the results are often available as printed maps or written descriptions. In order to analyze language spatiality, the information must be digitized into geospatial data, which contains location, time and other parameters. When compiled and harmonized, this data can be used to study changes in languages' distribution, and combined with, for example, population and environmental data. We also utilized the knowledge of language experts to adjust previous and new information of language distributions into state-of-the-art maps. The extensive database, including the distribution datasets and detailed map visualizations of the Uralic languages are introduced alongside this article, and they are freely available.

**Data Availability Statement:** Geographical database of the Uralic Languages (S2 Appendix) is available from http://doi.org/10.5281/zenodo.

## Introduction

Language geography has recently gained new attention from the growing interest in human history research, which draws evidence from genetic, cultural, and linguistic studies [1–4]. The data from different disciplines studying the human past often includes spatial and/or temporal dimensions, i.e. information about the location and time of each observation. These parameters can be utilized when fusing diverse data on human history as spatial information. Although the usage of geographical distances, for example in gene–language correlation studies, has become more common since a seminal paper by Creanza et al. [5], spatial data and methods have much untapped potential in studies of human history.

Linguistic research has often concentrated on non-spatial aspects [6], and geographical inventories grew in number as late as in the 19th century [7]. Many of the first maps depicting the distribution of languages (often labeled as language area or speaker area) were actually illustrations of the locations of different ethnic groups [8–10], at a time when ethnic and

4784188. All other relevant data are within the paper and its Supporting Information files.

**Funding:** TR was financially supported by the University of Turku Graduate School (UTU-BGG); https://www.utu.fi/en/research/utugs/doctoral-programme-in-biology-geography-and-geology), Kone Foundation (UraLex and personal grant); https://koneensaatio.fi/en/grants/, Finno-Ugrian Society; https://www.sgr.fi/en/, and UIT – The Arctic University of Norway; https://en.uit.no/startsida. OV was funded by Kone Foundation (SumuraSyyni, AikaSyyni); https://koneensaatio.fi/en/grants/, and the Academy of Finland, grant number 329257; https://www.aka.fi/en/. MR was funded by the Academy of Finland, grant number 329257; https://www.aka.fi/en/. The funders had no role in study design, data collection and analysis, decision to publish, or preparation of the manuscript.

**Competing interests:** The authors have declared that no competing interests exist.

linguistic identity were strongly connected. Throughout the history of linguistic cartography, language distributions have often been presented on published maps as non-overlapping regions, or simply as a text label over an approximate location. Occasionally, the distribution of languages has been documented only in written text, especially at the dialectal level. For the sake of cartographic clarity, language maps have also been commonly created from a monolingual perspective or using political mapping units concurrently concealing the regional diversity of languages [11, 12]. The spatial accuracy of the location information in the original studies varies greatly, as some sources aim at giving an overview of the whole language family, whereas others provide a detailed view of individual languages or dialects. In addition, a systematic description of the original mapping methods is often lacking, which complicates the comparability of the data sources.

There are about 7000 languages in the world [13, 14], and except for language isolates, none of the language families are comprehensively and uniformly represented as digital spatial data. Linguistic databases are often focused on linguistic (grammatical, lexical) data instead of exact non-linguistic data (the location of the speakers and speaker communities). Many online linguistic databases, such as Ethnologue [14], The World Atlas of Language Structures (WALS) [15] and Glottolog [16], contain general spatial information on languages' locations, branches (subgroups within a language family) and families as geographic points (geographic coordinates), but many of these services are not targeted to provide language areas (polygons) or to study their possible overlap.

Historical spatial language data is often diverse and scattered across analog publications that may even be difficult to find and obtain. We applied geographic information systems (GIS), which enable combining, analyzing and visualizing spatial data, in research on language geography, using Uralic-language areas as a case study. We introduced best practices for collecting and converting such data from the original sources into a harmonized and comparable digital form to create a spatial database of language distributions. This serves a wider purpose in the linguistic domain to promote data interoperability and sharing, with e.g. new standards for cross-linguistic data formats [17]. A geographical approach in linguistic studies has been promoted in several projects e.g. [1, 18–24], which utilize GIS to enable spatial visualization and easy updates.

The Uralic languages, spoken in Northwestern Eurasia, provide a compact case for developing a consistent methodology for the collection and harmonization of diverse spatial language data. The Uralic language family is one of the most studied language families, but it presents a less complicated case than, for example, the globally spread Indo-European language family. Depending on the linguistic (structural and sociological) criteria chosen, there are about 30–50 individual Uralic languages with a total of 20 million speakers [25]. Most of the languages are minority languages with only tens to tens of thousands of speakers on both sides of the Ural Mountains in the Russian Federation, while Hungarian, Finnish and Estonian are majority languages in their respective regions, having more than one million speakers each. In terms of size it is among the largest language families, but with around 40 languages, the amount of spatial information is still manageable, providing an excellent test case for compiling a database of language distributions for the whole family with uniform criteria. The spatial data of Uralic languages widens the recently published digital linguistic material on the Uralic basic vocabulary with cognate coding [26–28] and linguistic typology [29].

Our aim was to develop best practices for converting historical and current language-distribution information into digital spatial data, which is comparable to other spatial data and accessible to a wide audience. To achieve this, we compiled and published the first comprehensive spatial database of the Uralic languages. The ultimate goal was to promote the usage of spatial data in linguistic studies, as well as to improve opportunities for multidisciplinary

spatio-temporal research. The best practices cover different work phases from data compilation, digitization and harmonization to visualization and verification of language-distribution information through a structured expert evaluation process including also data sharing of the database as open data. In addition, to illustrate the state-of-the-art on the historical geography of the Uralic language family, we created and published a comprehensive collection of historical and current maps based on the datasets. The data and maps are freely available in the Zenodo data repository and *Uralic Historical Atlas* (URHIA) under a Creative Commons license.

## Methods

### Methodological considerations

The amount and quality of spatial information of the languages vary between the language families. Instead of digital data, the distribution of the languages is often available only on analog maps and text sources. To be able to use spatial language data in new map visualizations or research with other spatial linguistic or historical datasets, the information needs to be transformed to the digital format, and in addition to be stored in the same database.

Determination of a language distribution is complex. Without a unified method for defining languages on the map, the process includes many subjective cartographical and linguistic elements such as how to take into account variations in population density, ethnic groups' mobility within their living environment and occurrence of bi- or multilingualism as well as the very definition of a language itself on one hand, and the speakers of the language on the other. The lack of systematic description of used mapping methods also complicates the comparability of different historical source materials. However, the development of the actual standard for language area is beyond the scope of this work, and the distributions of the languages are presented exactly as they were defined in the original publications, i.e. the spatial extent of languages remain unchanged in our process. In addition, structured expert evaluations are used to reduce the existing uncertainties in the original publications as well as to increase the harmony between the past and present information of language distributions.

In the following chapters we introduce the developed guideline on how heterogeneous spatial language data can be converted to consistent geospatial data by taking into account the standards of linguistics and geographic information science. The workflow consists of 'Data collection and harmonization', 'Creating state-of-the-art maps based on the digitized data and new expert opinions', and 'Aspects of data sharing and licensing'. The Uralic language family works as a test case in this study, but the methodological guideline to create consistent geospatial data and database also applies to languages spoken in other geographical regions.

### Data collection and harmonization

The existing information about past and present distributions of languages are seldom available as digital spatial data. This was also the case with the Uralic languages for which most of the spatial information was available only as printed maps and text descriptions published since the end of the 19th century, starting from Donner [30, 31]. In addition, information on language distribution was scattered across numerous publications, and the mapping methods used in these studies were highly variable. For example, some studies presented the geographical distribution of the whole language family e.g. [30–33], while others concentrated on individual branches e.g. [34] for (Ob-)Ugric, [35] for Permic and Ugric, [36] for Saami and [37] for Finnic. The pioneering map by Donner [30] did not include the Samoyedic branch, which at the time was not unanimously considered a part of the Uralic family. Donner used the terms Finno-Ugric and Uralic synonymously, whereas the subsequent tradition has often regarded

Uralic as consisting of Samoyedic and the remaining Finno-Ugric languages. Spatial language information was most often published in individual language maps with dialect divisions, which were the most detailed mapped information.

The spatial accuracy between different languages varied because of the different amounts of available information at the time of the original studies. However, the spatial information of languages was most commonly represented as areas on the maps. To be able to make uniform and comparable representations of different language distributions we visualized those as areas in GIS. In practice, we digitized the data as vector polygons (closed areas including the boundaries making up the areas) instead of points, lines or raster surfaces, which are other options to visualize where languages are spoken on the digital map. The use of polygons allows the presentation of the exact shape and location of the objects depicting the language distributions. However, in cases where spatial information of languages needs to be presented in a more general level or the occurrence of language is point-like such as one village, the usage of point geometry type can be equally reasonable.

The basis of the digitization process was to define each language area as precisely as possible while avoiding overly detailed information in the map visualizations. There were several sources for each language where geographical distribution was provided as analog maps. In many cases, the opinions of the exact location and spatial extent of a language varied between the sources. Thus, we compiled different distributions from languages, covering 1–8 sources per language.

We collected the information concerning the time period before the extensive changes in Uralic language areas during the 20th century. Therefore, the mapping distribution approximately depicted the situation at the beginning of the 20th century, which is seen as the maximum distribution of the Uralic languages in general. This period is labeled as traditional. For the Sayan Samoyedic languages (Kamas, Mator), which became extinct in the 20th century at the latest [38], the traditional distribution refers to the beginning of the 19th century. We also collected the language distributions corresponding to the current situation, covering approximately the first two decades of the 21st century. The current geographical distribution of the languages was collected using the same principles of spatial generality and accuracy as with the past distributions. This decision ensures the comparability of the data from different time periods, and enables their use in map visualizations and spatial analysis.

The original spatial information was transformed into geospatial data using consistent methods. First, the original maps were scanned and saved in a digital image format suitable for GIS software (see more detailed description of the digitization process in e.g. [39, 40]). Second, the scanned and electronic language maps were georeferenced, i.e. tied to a geographic coordinate system using reference basemaps (such as Open Street Map, Google Maps), and properly selected ground control points. As a coordinate system we used the World Geodetic System 1984 (WGS84), since it is a widely-used standard coordinate system for global and regional level data (on average larger than nationwide geographical area), also in linguistic databases such as Glottolog and WALS. Third, the language distributions were digitized into vector shapefiles (shp), i.e. the geographic information was created as georeferenced polygon objects from the maps (language area was determined exactly as in the original publication). At this point, the text descriptions of the language distributions were also digitized into polygon objects. In some rare cases, especially at the dialectal level, text descriptions were the only information available of distribution, and it should be noted that the transformation from written descriptions into polygons is more vague and subjective than digitizing printed maps.

After processing the spatial component of the source data, we added the ID, name of the language and dialect, and names of the branches they belong to, together with an indication of the time period that the language distribution corresponds to (Table 1). We also included

**Table 1. Recommendations for the suitable contents of the geospatial datasets presenting the distribution of languages including the benefits of each, and our solutions (selected in the case study) concerning the Uralic languages.**

| Character | Advisable types/features | Benefit/comment | Selected in the case study |
|---|---|---|---|
| Data type | Vector data | Enables the exact location and shape of the object | Vector data |
| Geometry type | Polygon, point | Polygon: Works for areal data, presenting the object's boundaries | Polygon whenever possible, point in few exceptions |
| | | Point: Works for presenting the point-like distribution or extensive distribution in general | |
| File format | Interoperable, up-to-date format, e.g. SHP, GEOJSON, WKT | SHP: Widely used, easy to convert | SHP |
| | | GEOJSON, WKT: open source-based, new technology | |
| Coordinate system | WGS84 | Standard in digital map services, works for global and continental-wide data, compatible with other spatio-linguistic and interdisciplinary data | WGS84 |
| Attribute data | ID/FID, language, dialect, branch, time period, sources, Glottocode, ISO code, other information | Increases information on the identity, usability and sharing | All suggested |
| Temporal divisioning | Data-specific, e.g. exact date, division by centuries or more general approach when appropriate | Exact date: When dating is well-known (present-day data) | More general: Division to traditional–current |
| | | Division by centuries: Well-known historical data | |
| | | More general divisioning: imprecise historical data | |
| Metadata description and file naming | Comprehensive description of data content including at least: file format, data type, coordinate system, data sources, temporal extent and ownership; executed with a logical file naming | Enhance data's systematicity, transparency and usability | All suggested and point of contacts, maintenance frequency |

Geospatial data consist of spatial (location: coordinates, place name, etc.) and attribute data (features: name or ID of language, name or ID of dialect, etc.). To achieve the best possible structure and operability for each datasets, a data-specific approach is recommended.

references to the original source(s) in order to distinguish between different source materials. We also included the respective language's Glottocode (language ID produced by Glottolog) and ISO 639-3-code (another ID for languages produced by the International Organization for Standardization) within the attribute table. Glottocodes and ISO codes were developed for identifying languages, and they can be used, for example, for identification in cases where languages have several alternative names.

We created the geospatial data to be compatible with the existing linguistic data, as well as with data from other disciplines. We therefore aimed to utilize the data formats and practices previously used in research into human history. To make the data findable, trackable and transparent, and to improve the data's usability, we paid special attention to describing the contents of the data. The content description, i.e. the metadata, provides information about e.g. the file format, data type, data sources, coordinate system, temporal extent, point of contacts, ownership, metadata author and maintenance frequency. The metadata management plan also focused on the systematic naming of the dataset files in the catalog (naming conventions for the filesystem directories that hold the data), which is especially important when there are several distributions for one language. Consistent naming facilitates computer-aided search and provides information about a dataset file's contents without opening the dataset file itself. In addition, the datasets within the database are structured based on the general linguistic classifications of the Uralic languages.

In general, when creating the geospatial data to serve a wide range of users it is not justified to limit data feature options strictly. The selection of different solutions during the data creation should be data-dependent, but also the diversity of the end users (GIS vs. computational users) and their expected different working methods can be taken into consideration.

Therefore, we decided to utilize flexibility when recommending the different practices for geospatial data creation and harmonization (Table 1). For example, in a case of file format selection the recommendation is to emphasize interoperability and convertibility, for which there can be several suitable formats. Concerning the spatial representation of language distribution the usage of polygons should be the primary option even though some limitations in the amount and quality of spatio-linguistic data advocate using points alongside the polygons. In the historical context, the exact dates of data are not often realistically achievable, especially when going further back to history. Therefore, temporal divisioning should be done as precisely as possible, but in many cases more general division can be preferred to achieve consistent spatio-temporal datasets. In conclusion, systematic implementation through spatial linguistic data processing with comprehensive descriptions of the data contents is crucial when targeting the harmonized geospatial data.

## Creating state-of-the-art maps based on the digitized data and new expert opinions

After harmonizing digital spatial data as coherent geospatial datasets they can further be used in map visualizations and spatial analysis. For example, further comparisons of geographical extents from different sources are easy to execute by overlaying separate layers in GIS. The possibility to visualize several layers simultaneously on the map enables a visual inspection of how the language boundaries have been drawn in different sources. It also allows the creation of updated language distributions and thereby improved language maps based on all collected data and basemap features (e.g. information about land and water areas, topography, other natural environment, settlements and administrative boundaries, as well as place names) relevant to understanding the geographical context of a particular language. An updated map visualization can be based on one source depicting the geographical distribution of a language, or the use of several sources. The reliance on only one language extent is straightforward in cases where the distribution of a language is unambiguous. However, in many cases, the geographical distribution of a language is not unambiguous, as different sources present spatially variable views of the distribution (Fig 1). Thus, a new, optimized distribution map of the particular language can be created by examining the different overlapping layers simultaneously, and creating criteria where different characteristics are weighted (see e.g. [24]). For example, a new distribution for a language can be delimited using the common extent occurring in all source materials, and leaving out the areas that occur only in some of the sources. The features can also be prioritized related to the original mapping method, spatial accuracy or reliability. The novelty of the original sources can also be one of the factors regarding the determination of the new distribution of a language.

In our case, it was obvious that different opinions about the language distributions vary notably between the different sources by language. On the other hand, information about the present-day distributions was insufficient. To be able to create spatially consistent state-of-the-art maps for the past and present distributions, we developed a structured expert evaluation process instead of examining the geographical distribution presented in original sources by ourselves. This methodology is particularly applicable for well-known language families, which are being actively investigated. In practice, we utilized a comprehensive database of compiled language distributions. We also collaborated with professional Uralic linguists in the process in order to gain new spatial knowledge on the individual languages, which was unavailable in existing published material. The utilization of expert reviews was useful also because they included an assessment of the previously produced material and evaluated its accuracy in relation to new information.

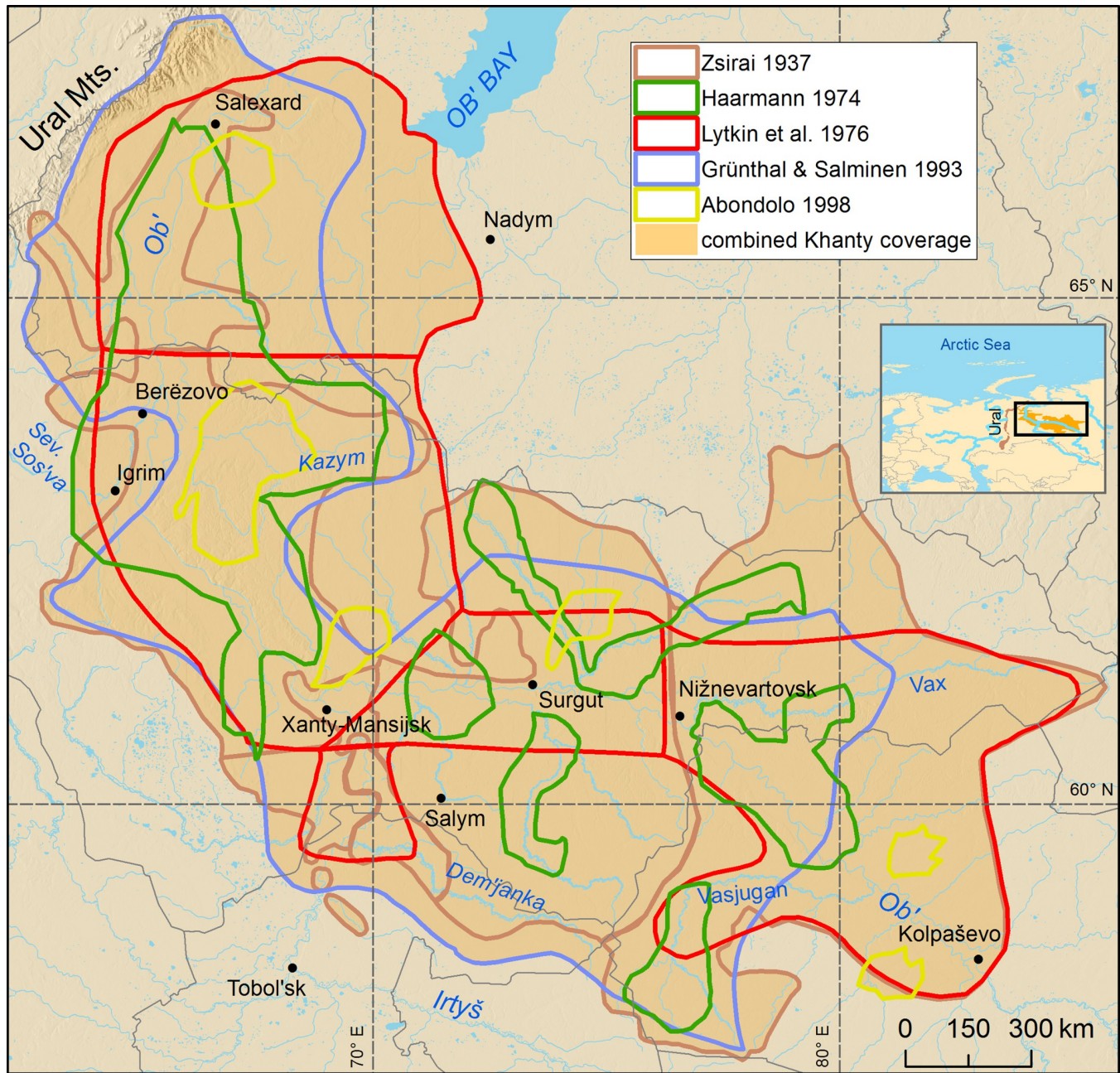

**Fig 1. Geographical overlap of different source materials concerning the distribution of the Khanty language(s) at the beginning of the 20th century.**
Original sources Zsirai [34], Haarmann [41], Lytkin et al. [35], Grünthal & Salminen [33] and Abondolo [42] have been visualized using boundaries of each polygon. A solid green area has been created merging the distributions of all Khanty sources, and it is indicating the area where Khanty could have been spoken. Basemap datasets from Natural Earth [43], Digital Chart of the World [44] and ESRI [45].

First, we visualized all distributions of each language on draft maps. Second, we designed a query, including the output of visualizations and a set of customized questions to gather structured expert knowledge about each of the Uralic languages (see a more detailed explanation in Rantanen et al. [46], S1 Appendix). The experts consisted of the authors of *The Oxford Guide to the Uralic Languages* (2022) [47], the most comprehensive handbook of the Uralic family ever produced. Each expert or group of experts (in cases where responsibility of a particular

language chapter was shared between more than one author) provided a consensus opinion on draft map regarding the language of their expertise. They were queried about which of the original sources correspond most precisely to their understanding of the language distributions at the beginning of the 20th century, and if none of the sources agreed with the current understanding, where and how the boundaries should be edited (S1 Appendix). We also inquired about the 21st-century distributions of the languages, which is information that was almost totally missing on the preexisting maps. Simultaneously we inquired about the relevant place names (settlements, administrative units, water bodies, natural environment) in the correct spelling to put on the map. Because the queries were assigned only to the responsible author(s) of a particular language chapter, we avoided the possible inconsistencies the language experts may have on the distribution of the languages. In a way, the pool of experts was a preexisting natural set of specialists who had been selected by the handbook editors before the cooperation project. These about 30 experts, in turn, consulted dozens of other specialists and speakers of the languages of their expertise.

The expert survey yielded a significant amount of new information concerning the past and present distributions of the Uralic languages, and created an excellent basis for the production of the new state-of-the-art Uralic language maps. All state-of-the-art Uralic language maps were complemented by the expert evaluations, but the amount of new information varies among the languages and time periods. In some cases, the presented past distributions strictly followed earlier studies, but in others there were notable changes. The information of the current distributions were received almost as a whole from the experts, and as an exception for the overall usage of polygon type, it was reasonable to use points alongside with polygons in some map visualizations. In sum, new distributions for all languages were determined in accordance with the opinion of the experts. The sources that were used to create a new distribution for the languages are comprehensively presented in figure captions.

For this publication, we created three types of visualizations: 1) individual language maps, 2) maps for the main branches of the Uralic languages, and 3) an overall map of the whole language family. The maps present the most recent and precise information on the geographical distribution of each Uralic language. All maps in each category were based on the same datasets, but the most detailed information, including dialect areas, was usually presented in individual language maps. To achieve visual consistency and clarity among the collection of maps, we decided not to present overlapping areas of different languages. At the same time, we did not indicate the areas of bilingualism or multilingualism on the map, even though bi- and multilingualism commonly occur in the overlapping areas. Suitable accuracy and spatial scale were selected separately according to the purpose of each map.

We also provided the created map drafts to *The Oxford Guide to the Uralic Languages* [47] in return. The visually modified versions of the maps presented here are published there alongside each text chapter, which serve to introduce the Uralic languages.

## Aspects of data sharing and licensing

Spatial data platforms play an important role in making it easier for users to publish and access scientific geospatial information. To maximize the accessibility of the Uralic language spatial data, we first stored all the compiled and harmonized datasets (shp) and map visualizations as images (png) in the same spatial database called the *Geographical database of the Uralic Languages* [48]. Then all the data were stored in the Zenodo data repository, and published under the Creative Commons CC BY 4.0 license, allowing flexible possibilities to manage the data (e.g. data uploads without waiting time, as well as usage statistics and DOI (Digital Object Identifier) citation). The permanent DOI link enables effortless citation of the data and

eliminates the problem of ever-changing web addresses. Whenever it is necessary to edit or update the uploaded dataset files, Zenodo registers every new version number (e.g. v.1.0., v.1.1), so that it is also possible to track the evolution of the database.

All human history researchers or lay audiences can not be expected to master geospatial techniques [49, 50]. Therefore, the full benefits of the published database can be difficult to achieve. For example, to be able to create own map visualizations based on the datasets, a basic understanding of the usage of desktop GIS is required. To serve especially the audience who are not fluent GIS users, we published the new Uralic language maps in Zenodo as images in PNG format. The database with the datasets and maps are available also in the *Uralic Historical Atlas* (URHIA) [51], which is an interactive spatial data platform with a map view [52], enabling visual inspection in a web browser without the need to download the datasets. The URHIA map interface also enables the creation of own customized map visualizations and serves a possibility for loading them as multiple different file formats such as SHP, CSV or GEOJSON.

## Results

### Practices for spatial language data harmonization, visualization and sharing

To improve the opportunities to carry out spatial historical research from linguistic and inter-disciplinary perspectives, we introduce a methodological guideline for unifying and presenting the geographical information of language distributions (Fig 2). We operated in the context of the Uralic language family, but the workflow is applicable to other language families or

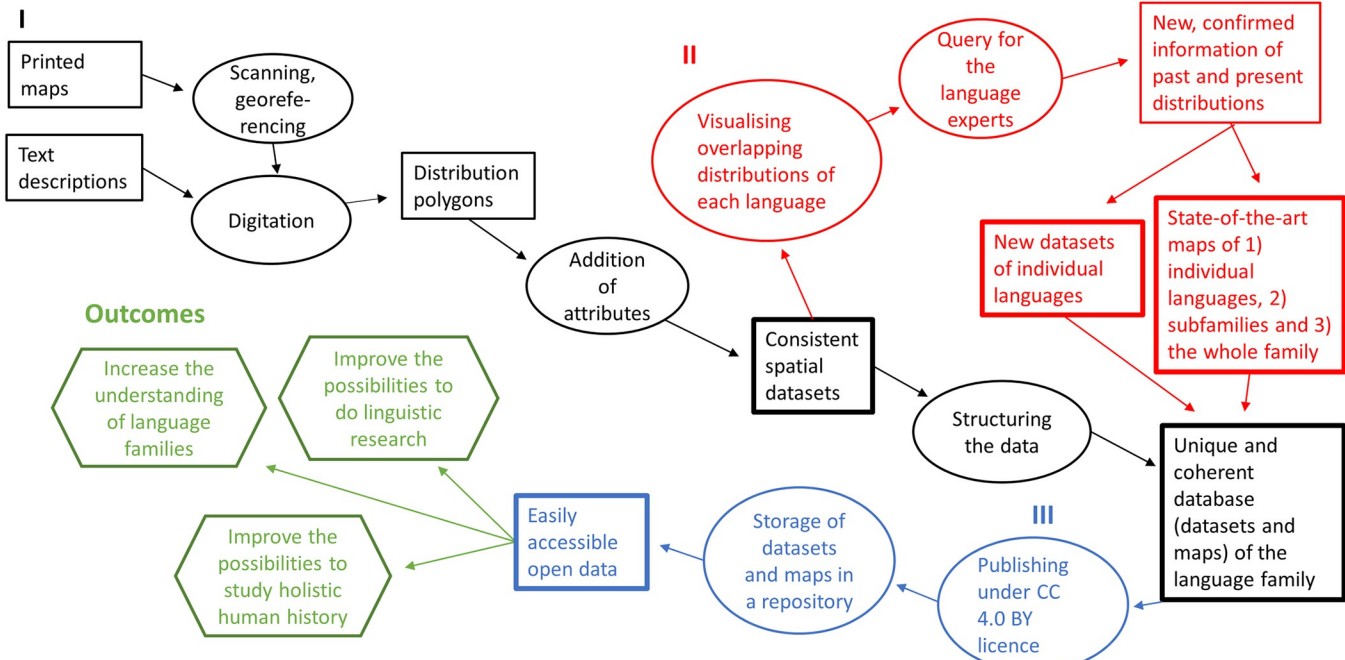

**Fig 2. Workflow for best practices in handling of language family data includes three separate phases: I processing and harmonization of spatial data collection: A path from analog and digital source data to a consistent geospatial database, II visualization combined with queries from experts in the case of lesser-studied languages, and creation of improved new maps based on updated information, III data sharing.** The outcomes of the best practices increase research opportunities and general understanding of language distributions. Original data and output are shown as rectangles, processing as ovals and overall benefits as hexagons. Details of the workflow are described in Section 'Methods'.

geographical areas as well. As a result, we suggest a three-step process, using the Uralic language data to exemplify the workflow: I) all the spatial source material is digitized into geospatial data using a systematized procedure for data collection, where spatial and attribute data is processed into a comparable and consistent form, which is stored in a database with uniform settings; II) the language distribution data is verified by experts in the particular languages, resulting in new and updated information on past and current language distributions, and state-of-the-art maps are created based on the expert review; and III) open data sharing ensures the usability of datasets in research. It should be noted that a three-step process can be used to digitalize, harmonize and upgrade all kinds of historical spatial data from diverse analogical sources. The developed guideline can be applied also without step II (the expert evaluation) in cases where a particular language has no experts to evaluate the distribution based on different presented opinions. In these cases, some other well-reasoned method to generate state-of-the-art distributions should be used (different options are presented in 'Creating state-of-the-art maps based on the digitized data and new expert opinions').

## Geographical database of the Uralic languages–geospatial datasets

*The Geographical database of the Uralic languages* [48], published in Zenodo (S2 Appendix) covers the geographical distribution of all Uralic languages (S1 and S2 Tables) in roughly two time periods: 1) at the beginning of the 20th century–indicating approximately the widest known distribution of Uralic languages, labeled traditional in what follows, and 2) a current distribution covering approximately the beginning of the 21st century up to the present day. There are 1–8 traditional and 0–2 current distributions available for each language, compiled initially from published sources and secondarily updated and improved by experts in these languages (S1 Table). The database follows a hierarchical structure presenting both the individual branches of the family (e.g. Saami, Finnic, Samoyedic), the individual languages within those branches (e.g. Saami: North Saami, Skolt Saami, Kildin Saami), and some dialectal divisions within individual languages (e.g. North Saami: Torne, Western Inland, Eastern Inland, Sea). Note that the hierarchical structure of these languages takes no position on how to taxonomically position the (uncontroversial) branches within the family or individual languages within the branches they belong to.

The total number of datasets is 226 (Table 2). Each dataset consists of the spatial location of the language either polygons, which is principally selected geometry type (222 cases) or data points, used in few well-reasoned exceptions (4 cases). All datasets are available as shapefiles in the WGS84 coordinate system. The attribute information consists of the FID (feature ID), language/dialect name, information on the branch it belongs to, the time period, original sources, Glottocode (language ID) and ISO 639-3-code (another ID), all according to international linguistic standards. The language IDs allow merging the datasets with existing language data operating with the same codes. By constructing the datasets uniformly, the usability of data is optimized also with other kinds of spatial data, such as D-Place [53], which provides a vast amount of cultural and environmental information. In addition, metadata descriptions that

**Table 2. The number of dataset files divided into the original published studies (original) and expert-modified distributions (expert) with two overall time periods.**

| Time period | Original | Expert | Sum |
|---|---|---|---|
| Traditional | 148 | 55 | 203 |
| Current | 3 | 20 | 23 |
| Sum | 151 | 75 | 226 |

introduce the data collection methods and the data characteristics were comprehensively created.

## Geographical database of the Uralic languages–state-of-the-art language maps

In addition to the geospatial data, the database (S2 Appendix) presented here consists of 45 maps with colors depicting the location of past and present distributions of the Uralic languages (S2 Table). The maps are divided into the following categories, each of which is illustrated in this paper as example maps, which introduce the hierarchical structure of the database and the temporal dimension: 1) an overall map of the whole language family (Fig 3), 2) maps for the nine uncontroversial main branches of Uralic (Fig 4), and 3) individual language maps (Fig 5). All map levels are based on the same original source data, but the most detailed information exists in the individual language maps (Fig 5), which predominantly also includes dialect distributions and thus forms the optional fourth level in the hierarchy. In some exemplary cases, past and present distributions are shown as their own layers on the map (see examples in S2 Appendix), but in some cases, there are separate panels for the time periods (Fig 5A and 5B). The layout of each map has been customized independently, emphasizing the environmental (lakes, rivers, topography), cultural (settlements, nomenclature) and

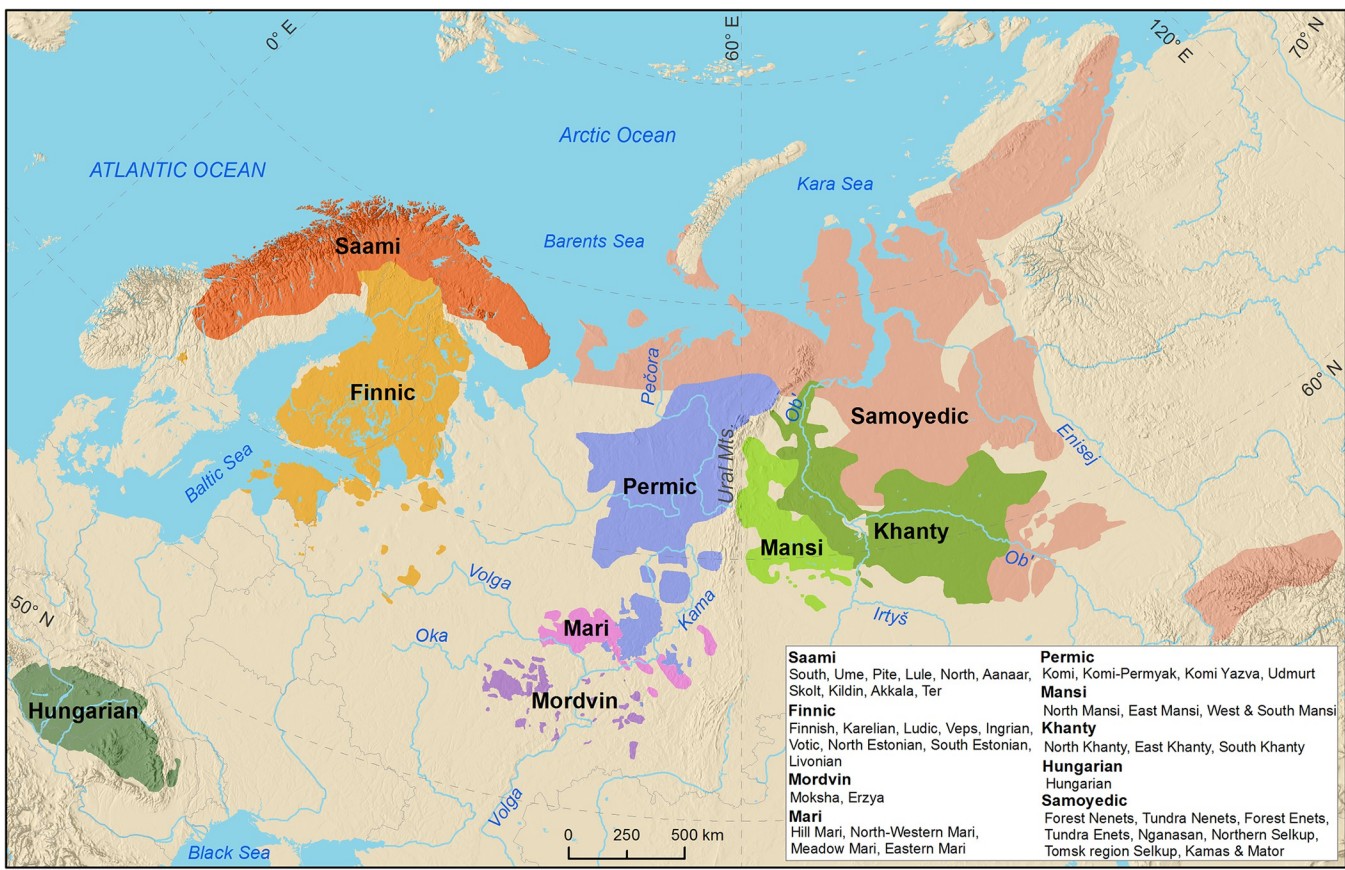

**Fig 3. Geographical distribution of the Uralic languages at the beginning of the 20th century.** The uncontroversial branches of the family are presented without overlapping areas. A list of original sources is available in S2 Appendix. Basemap datasets from Natural Earth [43], Digital Chart of the World [44] and ESRI [45].

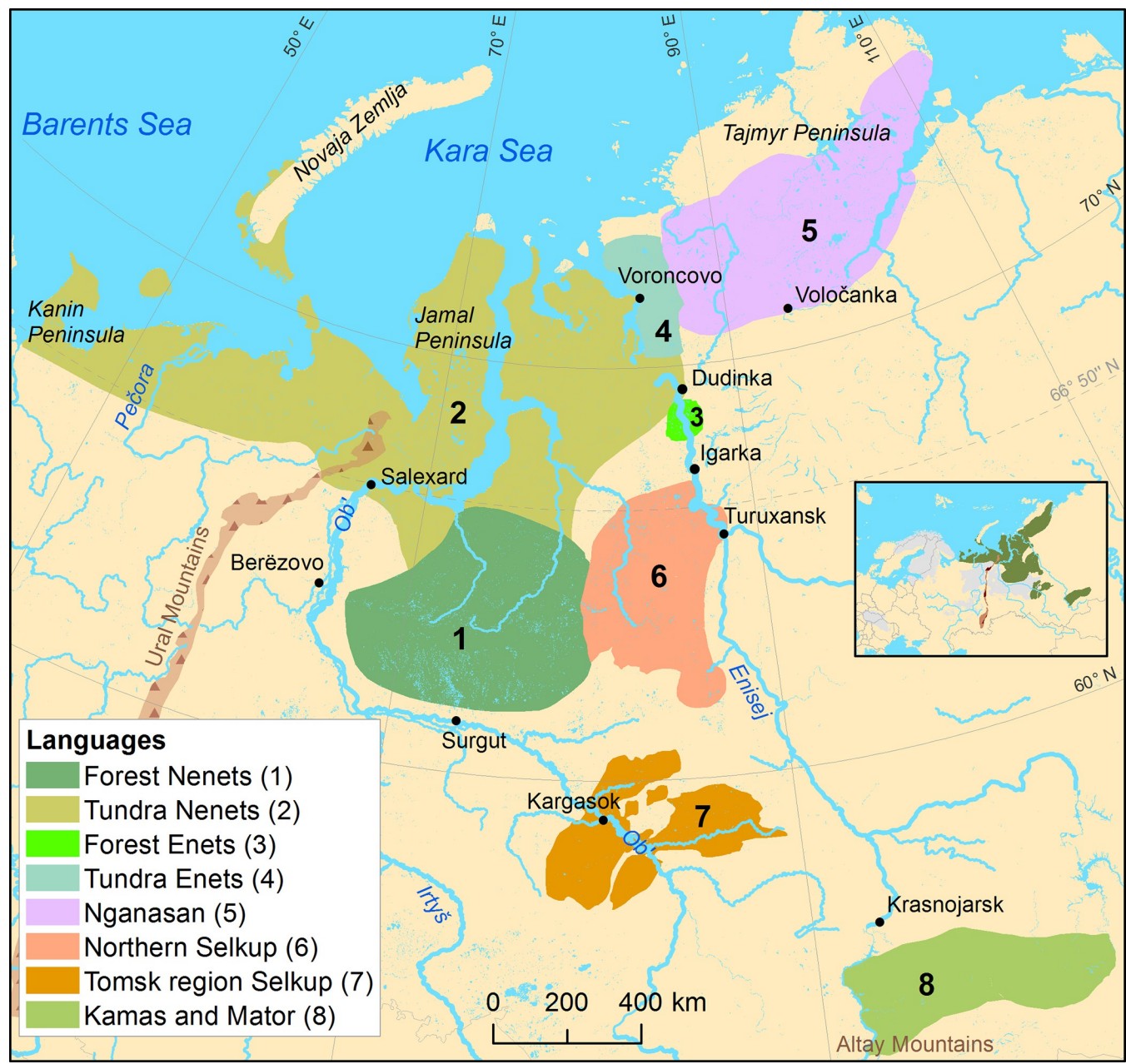

**Fig 4. Samoyedic languages at the beginning of the 20th century.** Languages are presented without overlapping areas. Original sources: Soviet Census of 1926 [54], Popov [55], Dolgikh [38], Dolgikh & Fajnberg [56], Dolgikh [57], Verbov [58], Grünthal & Salminen [33], Helimski [59], Tuchkova et al. [60], Siegl [61], Brykina & Gusev [62]. Basemap datasets from Natural Earth [43] and Digital Chart of the World [44].

political features (administrative borders) which facilitate an understanding of the spatial context of a particular language. To achieve visually clear and easily comprehensible illustrations, overlapping languages are not shown on the maps.

## Discussion

Best practices for the processing of spatial language data were developed in the course of digitization, harmonization and sharing of cartography on the distribution of the Uralic languages.

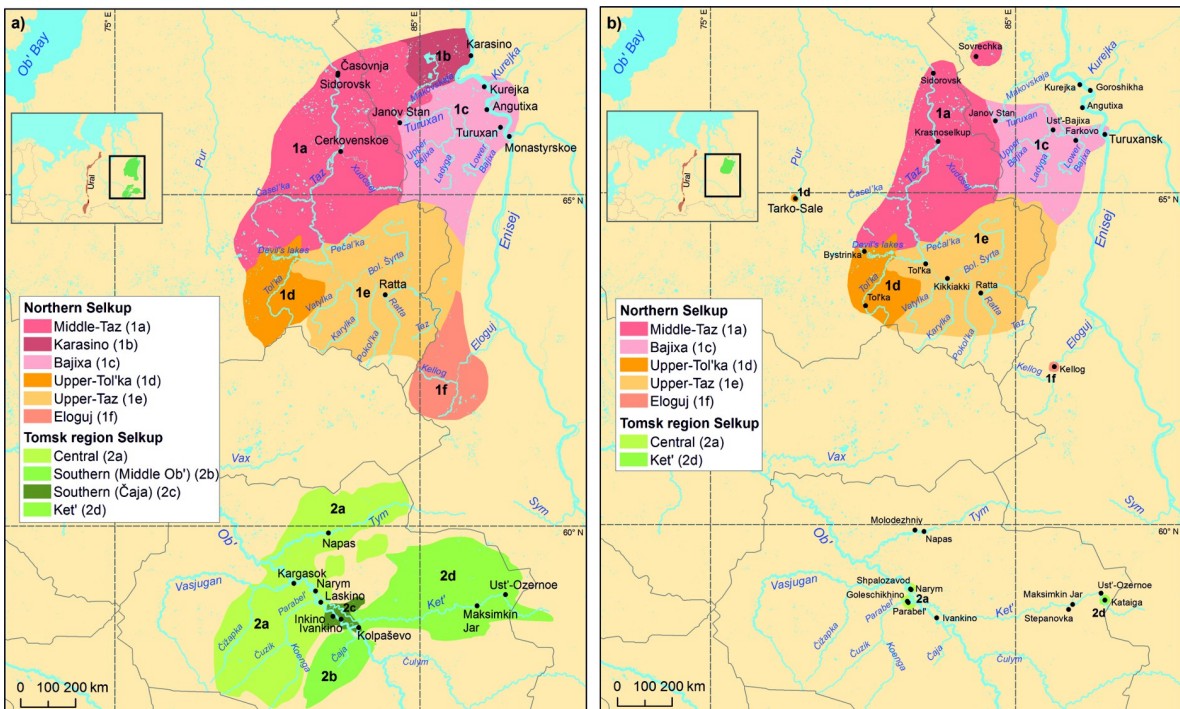

**Fig 5.** Traditional (a) and current (b) distribution of Selkup. A comparison of the maps demonstrates the changes in language and dialectal distribution over time. Original sources for traditional distribution are Grünthal & Salminen [33], Tuchkova et al. [60] and for current distribution Tuchkova et al. [60], Kazakevich [63]. Basemap datasets from Natural Earth [43] and Digital Chart of the World [44].

However, the suggested process is applicable to other current and historical spatial data, including other areas and language families, as well as data from other disciplines, such as archaeology and genetics. The benefits of consistent practices are apparent: the language distribution data created is coherent and comparable to other geospatial data, and the data is uniformly described. The data is stored in one database, allowing customized map visualizations, visual comparisons, and further spatial analyses of the linguistic data, such as phylogeographical modeling of language spread. The availability of the data is secured through open-access publication of the *Geographical database of the Uralic languages* (S2 Appendix). The database includes finalized state-of-the-art maps for each language, and therefore it is not necessary to master GIS methods to use the geospatial information. We also offer easy access to map processing via spatial data platform URHIA [51].

Bringing the data on language distribution into the digital realm not only enables a review of the massive amount of work done so far in historical linguistics, but also opens new horizons for bridging the knowledge to linguistic research and teaching in general, as well as to interdisciplinary holistic studies of human history. The geographical approach allows location-based studies of language areas (which are increasingly desired) in parallel with, for example, archaeological, genetic and environmental data [64–66]. It must be noted, however, that the accuracy of language-distribution information is higher for modern times than for historical or prehistorical eras. It must also be noted that languages' distribution may have changed significantly; for example, the Saami languages have been present in most of Lapland for less than 1500 years [67]. The temporal dynamics of the language distributions are reflected in the data as time layers, as far as the original sources allow. Even though the time frame of the

documented Uralic-language distribution data does not extend far back in history, the temporal dimensions provide insight into the spatio-temporal dynamics of these languages.

In the context of the Uralic languages, the original yet sometimes unsubstantiated representations of language distributions have often been accepted as such, and the presented distribution boundaries have been perpetuated in maps to the present day. This basic setting affects the possibilities to create sophisticated visualizations of a historical language area in digital cartography. However, the conversion of historical data into digital spatial data, operating with polygons and points, remarkably improves the possibilities to use language-distribution data innovatively, for example by simultaneously visualizing multiple map layers. Using separate layers for comparing different original sources expands the possibilities to create new information on past distributions, which were previously not presented on maps.

In many cases, also in the history of the Uralic languages, different interpretations of languages' distributions at the same time periods have been presented by different authors. In the process of creating the Uralic languages' distributions as geospatial data, we turned to expert opinions in order to calibrate and harmonize the source data. This was to assure, for example, that the relationship between the source data and other knowledge concerning the population and cultural history of the region is accounted for. The decisions made by the original investigators, the descriptions of their methods, the geographical scale, and temporal coverage all have an impact on the data itself, but a careful expert review helps to unify these factors to a degree.

The main challenge concerning the mapping of language distribution, in general, is related to the definition of a language area. There is no established standard to determine the distribution of a language on the map [11, 46, 68–70], i.e. where the boundaries of language distributions should be drawn. Mapping methods have varied among the inventories, for example, according to the amount of existing data and the ultimate purpose of the map. Also, personal preferences may affect the visualization output, even though maps should be neutral and realistic [70]. Using a structured expert-evaluation process during the digitization of the source material is a feasible way to mitigate and adapt to the issue of how a language area is defined.

Also other issues, such as the uncertain definition of a speaker, difficulties in distinguishing between dialects and languages, variation in ethnic groups' mobility within their living environment (sedentary vs. nomadic lifestyle), regionally unevenly distributed populations and migration to new territories have complicated the interpretation of the geographical extent of the languages and emphasized the subjectivity of depicting language-distribution boundaries through history. In addition, systematic descriptions of the chosen methods are often lacking in the historical sources, making it challenging to assess the source data's quality and repeat the original methodology. Luebbering [71] presents an illustrative list of caveats that customarily accompany language maps. For further discussion about the history, challenges and suggestions for future work concerning the mapping of languages' distribution, see e.g. [46, 70, 72–74]. Our solutions to these challenges in the case of Uralic languages are documented in 'Methods'.

A common challenge faced when illustrating language distributions is that often several languages are spoken in one region, or even within one population. When presenting languages and dialects as individual objects in spatial data, this is not a problem, since overlap can easily be analyzed and visualized in GIS. Therefore, there is no need to stick to the classical cartographic representation of regional monolingualism, and we have also created each language distribution polygon of the Uralic languages individually. Thus, any area can include as many languages in the data as needed, and the polygons can and do overlap where multiple languages have been observed. However, when multiple such data layers are visualized on the

same map, the possible overlaps need to be handled adequately by using a clear classification for overlap areas.

Time as one component of spatial language data allows for analyzing the dynamics in the development of languages and dialect areas. At the simplest, overlaying distribution maps of different time periods show how the distribution of one language has evolved during the known historical period (see Fig 5 for example, [21]). Combining this with e.g. environmental data, opens further possibilities to analyze the spatial interaction between the speaker populations' migrations and changes in the environment.

To our knowledge, this is the first time that an entire language family has been mapped and visualized as one harmonized database. The database, including the distribution maps for the Uralic languages, is available in Zenodo [48], and the data has also been published in the *Uralic Historical Atlas* (URHIA) [51], which enables online visualization of spatial data in a map interface, together with other data from the region.

## Supporting information

**S1 Appendix. A query of the North Khanty language.** Similar query was sent to each responsible author(s) of the particular language chapters of *The Oxford Guide to the Uralic Languages*. These expert evaluations were created to collect up-to-date information of past and present geographical distribution of Uralic languages.
(PDF)

**S2 Appendix.** Rantanen T, Vesakoski O, Ylikoski J, Tolvanen H. Geographical database of the Uralic languages. Version v1.0.; 2021. Database: Zenodo [Internet]. Available from: http://doi.org/10.5281/zenodo.4784188.
(DOCX)

**S1 Table. Number of distributions per language and time period in the geospatial datasets.** Language distributions are based on the published studies (original) and separate expert evaluations (expert) done in collaboration with the authors of *The Oxford Guide to the Uralic Languages*. Some original studies do not separate subgroups of language branches, which is the reason to use branch or general names in the 'Language' column (labelled in italics) in some cases, for example '*Mordvin*' or '*Khanty*'. Branches: Saami (I), Finnic (II), Mordvin (III), Mari (IV), Permic (V), Mansi (VI), Khanty (VII), Hungarian (VIII), Samoyedic (IX). *Skolt Saami: Distribution after resettlement in 1950s. **Livonian: Medieval and 1900s distributions.
(DOCX)

**S2 Table. List of maps containing information on each Uralic language.** Table shows how many dialects are presented per language as well as information of temporal coverage. Total number of language maps is 45. Main branches of Uralic languages are indicated as Roman numerals. *Meänkieli and Kven are seen here as separate languages, in S1 Table both instead belong to Finnish language. **Lule Saami is divided to three proper and three transitional dialects. ***Mordvin branch consists of five Erzya language dialects and four Moksha language dialects.
(DOCX)

## Acknowledgments

The authors express their gratitude to the experts who participated in the map review process (Ante Aikio, Marianne Bakró-Nagy, Svetlana Burkova, Svetlana Edygarova, Ulla-Maija Forsberg, Riho Grünthal, Arja Hamari, Markus Juutinen, Olga Kazakevich, István Kenesei, Gerson

Klumpp, Eino Koponen, Nikolay Kuznetsov, Johanna Laakso, Elena Markus, Matti Miestamo, Karl Pajusalu, Michael Rießler, Sirkka Saarinen, Anneli Sarhimaa, Zsófia Schön, Florian Siegl, Mária Sipos, Elena Skribnik, Taarna Valtonen and Beáta Wagner-Nagy), and to others who have contributed to the research process (Hannah Haynie, Sofia Koskela, Dmitry Kuznetsov, Luke Maurits, Tua Nylén, Juho Vehviläinen and the BEDLAN research group members). Christopher Culver kindly proofread the text.

## Author Contributions

**Conceptualization:** Timo Rantanen, Harri Tolvanen, Outi Vesakoski.

**Data curation:** Timo Rantanen, Meeli Roose.

**Formal analysis:** Timo Rantanen.

**Funding acquisition:** Harri Tolvanen, Jussi Ylikoski, Outi Vesakoski.

**Investigation:** Timo Rantanen.

**Methodology:** Timo Rantanen, Harri Tolvanen.

**Project administration:** Outi Vesakoski.

**Supervision:** Harri Tolvanen, Outi Vesakoski.

**Validation:** Timo Rantanen, Meeli Roose.

**Visualization:** Timo Rantanen.

**Writing – original draft:** Timo Rantanen, Harri Tolvanen, Meeli Roose, Jussi Ylikoski, Outi Vesakoski.

**Writing – review & editing:** Timo Rantanen, Harri Tolvanen, Meeli Roose, Jussi Ylikoski, Outi Vesakoski.

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
