## [Decision Letter · Decision Letter 0]

17 Jan 2022

PONE-D-21-40182Best practices for spatial language data harmonization, sharing and map creation – a case study of UralicPLOS ONE

Dear Dr. Rantanen,

Thank you for submitting your manuscript to PLOS ONE. After careful consideration, we feel that it has merit but does not fully meet PLOS ONE’s publication criteria as it currently stands. Therefore, we invite you to submit a revised version of the manuscript that addresses the points raised during the review process.

 Although both reviewers judge the revisions needed to be minor they address different aspects, so the combined amount of suggested revisions would be more than minor. Both reviewers point to some cases where a particular practice might be improved or where a suggestion is not perceived as being optimal. No doubt, there is a room for different opinions about what is a best practice, so more discussion of alternative could be added. And sometimes there is still room for improvement in your own practice--it may actually strengthen the paper to admit for this possibility where relevant.

We look forward to receiving your revised manuscript.

Kind regards,

Søren Wichmann, PhD

Academic Editor

PLOS ONE

Journal Requirements:

2. We note that Figures 1, 2, 4, 5, and 6 in your submission contain [map/satellite] images which may be copyrighted. All PLOS content is published under the Creative Commons Attribution License (CC BY 4.0), which means that the manuscript, images, and Supporting Information files will be freely available online, and any third party is permitted to access, download, copy, distribute, and use these materials in any way, even commercially, with proper attribution. For these reasons, we cannot publish previously copyrighted maps or satellite images created using proprietary data, such as Google software (Google Maps, Street View, and Earth). For more information, see our copyright guidelines: http://journals.plos.org/plosone/s/licenses-and-copyright.

a. You may seek permission from the original copyright holder of Figures 1, 2, 4, 5, and 6 to publish the content specifically under the CC BY 4.0 license.  

Reviewers' comments:

Reviewer's Responses to Questions

**Comments to the Author**

1. Is the manuscript technically sound, and do the data support the conclusions?

Reviewer #1: Yes

Reviewer #2: Yes

2. Has the statistical analysis been performed appropriately and rigorously? 

Reviewer #1: N/A

Reviewer #2: Yes

3. Have the authors made all data underlying the findings in their manuscript fully available?

Reviewer #1: Yes

Reviewer #2: Yes

4. Is the manuscript presented in an intelligible fashion and written in standard English?

Reviewer #1: Yes

Reviewer #2: Yes

5. Review Comments to the Author

Reviewer #1: The data and practice behind this paper are extensively and quite

well-described but lacks the most important part, namely what they

consider the "language area" (that is to be mapped). After noting this

problem (pp 19-20) they offer no definition nor or declaration of how

they have dealt with it for the data at hand, except to say they've

consulted experts. Unfortunately the paper also lacks any detail who

is to be considered an expert, with what instructions they have

consulted the experts and how they weed out inconsistencies across

experts. These are relevant questions. Presumably the experts are the

same kind of people who write papers about the same languages, and as

the authors duly note, they have "different opinions" and "vary

notably". For example, for Nganasan the maps show a continuous area

including the area to the coast east of Lake Taimyr and north of

Nordvik island with sources "Dolgix 1963, Popov 1948, Brykina & Gusev

2015, Wagner-Nagy 2018:3". None of these sources sanction this chunk

as Nganasan settled. In particular, the extremely detailed maps by

Dolgix (temporally differentiated) and Popov (summer-winter

differentiated) --- based on their own extensive fieldwork --- stop

just east of lake Tajmyr, not including the NE area up to the coast.

You could of course say that area might have been hunting grounds and

certainly no other ethnic group lay claim to it, but then the entire

northern Tajmyr could just as well be mapped as Nganasan as hunting

grounds (Golovnev 1999 says this explicitly). Also, following Popov,

the settlements were always along drainages, and the interior areas hunting

grounds, but the maps show continuous areas but not including *all* hunting

grounds. What is the actual intent? Another example is South Saami

whose northen dialect on the map includes Södra Tärna with the sources

"Hasselbrink 1981–1985, Rydving 2008: 360–361, Rydving 2016, Maja Lisa

Kappfjell & Jussi Ylikoski (p.c.)" but Rydving is explicitly arguing

for South Tärna to be counted as Ume Saami. What was the reasoning here

and what informaed was asked of or provided by Ylikoski? The bottom line

is, there are two choices: (i) Either the paper stays without a definition of

the intended areas to map and/or a methodology for using the experts, but

then the authors have not defined a "practice", let alone a "best practice",

and the title should be changed accordingly, or (ii) such a "best practice"

is defined and included in a revision. This is the only important point,

everything else is essentially fine and the resource as such is excellent.

Non-important issues:

* Rephrase: "In our case, it was obvious that different opinions about the language distributions vary notably between the different sources by language."

* "but the workflow is applicable to other language families or geographical areas as well"

There are no experts for every language available for most families and areas

so this workflow is not applicable in the same form for others.

* It would be nice of more detail (or a reference) were given on how to

best digitize printed maps

* Imperatorskoe Russkoe Geograficheskoe, Obshchestvo -> remove comma

* Imperatorskogo Russkogo Geograficheskogo Obshchestva -> why genitive?

* Some Russian sources are in Roman, some in Russian majuscule and some

in Russian minuscule. Harmonize.

Golovnev, Andrei V. (1999) The Nia (Nganasan). In Richard B. Lee & Richard Daly (eds.), The Cambridge Encyclopedia of Hunters and Gatherers, 166-169. Cambridge: Cambridge University Press.

Reviewer #2: The idea of publishing a best practice example of handling spatio-linguistic data is quite good and the relevance of this publication should not be underestimated. Good practice guidelines for data handling is available in all natural sciences and is also available in some disciplines of the humanities. A general good practice approach should be part of the digital humanities but also requires specific components for each topic. This article combines general aspects of handling geospatial data and specific aspects of linguistics. The current review focusses on the first part.

Though, the workflow employs rather general steps, it can be helpful for considering all steps and after all, too narrow standards are rather preventing the concept from being applied. In particular pointing at the importance of open licences (cc) and open platforms (zenodo) is an important consideration for up to date research. Another important point is to acknowledge certain redundancies such as offering the geospatial data as well as the ready made maps in order to allow benefits for different target groups.

The part on geospatial data can be further improved.

- It seems clear why polygons should be used, but why are four point data sets part of the database? It seems that this fact destroys the consistency of the database.

- I understand why WGS84 is used but for mapping and geospatial analysis projected coordinate systems are usually preferred. It would be helpful to explain the decision.

- The decision for shape files is not understandable at all. It is true that shape files are still wide spread but they are based on an completely outdated technology. A new god practice guideline and a new workflow should be based on up to date and sustainable technology. For small data as in this case, a text and WKT or EWKT based format such as csv would have the benefit of being easily to integrate in full reproducible research workflows. Further more, this formats are as software-independent as possible. For larger data spatialite databases and derivates such as geopackage would be a good choice. It would be helpful to discuss this point and to refer to newer developments in geography and geospatial technology.

- An example of overlapping data would be helpful and a more detailed explanation of how to combine data and handle the overlapping would be appreciated.

Another possible improvement could address the handling of chronological information. Perhaps, a more general approach would make sense. Categories such as "traditional" might make sense in a specific case but are rather confusing and imprecise in general. Examples of how to handle temporal data are known from temporal data bases and from archaeology.

6. PLOS authors have the option to publish the peer review history of their article (what does this mean?). If published, this will include your full peer review and any attached files.

Reviewer #1: **Yes: **Harald Hammarström

Reviewer #2: No

---

## [Author Response · Author response to Decision Letter 0]

25 Apr 2022

Reviewer #1: 

Comment: The data and practice behind this paper are extensively and quite well-described but lacks the most important part, namely what they consider the "language area" (that is to be mapped). After noting this problem (pp 19-20) they offer no definition nor or declaration of how they have dealt with it for the data at hand, except to say they've consulted experts. 

Response: This comment was valuable. The “language area” was not very clearly defined in the original manuscript. We made the following changes: an addition of a paragraph to the ‘Methodological considerations’ chapter, where the problematicity of determining the distribution of languages was described, including also the explanation of how language areas were presented in this work: “the languages are presented exactly as they were defined in the original publications, i.e. the spatial extent of languages remain unchanged in our process” (lines 122–125). In addition, we also referred to this aspect in ‘Discussion’ (lines 523–524) and added a reference to in-depth discussion in our chapter in The Oxford Guide to the Uralic languages (line 523).

C: Unfortunately the paper also lacks any details who is to be considered an expert, with what instructions they have consulted the experts and how they weed out inconsistencies across experts. These are relevant questions. Presumably the experts are the same kind of people who write papers about the same languages, and as the authors duly note, they have "different opinions" and "vary notably". 

R: The experts were indeed the researchers who wrote the particular language chapters to The Oxford Guide to the Uralic Languages. We now note this more clearly in lines 288–292. We also explain more detailedly how the expert evaluation process actually proceeded (changes in lines 276–279, 296, 299–304, 313–316, 319–320). To clarify the instructions we assigned to the experts, a new ‘S1 Appendix’ file was added to the manuscript. It includes an example of a query assigned to the North Khanty expert indicating the overall style of the all queries in this research. 

Concerning the comment on how we dealt with the inconsistencies across experts, we clarified the phrasing in the text (lines 299–304). In brief, there were no inconsistencies across different experts because we asked the opinion only from one expert or a group of experts representing the responsible authors of each language chapter of The Oxford Guide to the Uralic Languages. We also recognize that there are and will be different opinions about the exact boundaries of particular languages (at least as long as a standard for language mapping will be developed). In this case, the beginning of the process was to collect different opinions from published sources in the same database. Then, new state-of-the-art maps (consensus maps) were created on their basis. Experts evaluated the distribution based on these earlier sources as well as complementary information they had. After evaluating all the information simultaneously, the expert or (group of experts) made the decision on how to define the updated distribution on the map draft. These updated opinions ended up on the state-of-art maps as such. Further, updated opinions were also exported to the Geographical database of the Uralic languages, ‘S2 Appendix’, (former ‘S1 Appendix’) as ‘Expert distributions’ with ‘Original distributions’ (earlier sources). This process is now explained more detailedly in the manuscript (lines 275–316). Noteworthy, this is only one way to execute the consensus maps, and because all languages have no experts, we have explained the other relevant possibilities for creating certain kinds of maps in general level (lines 257–264).

C: For example, for Nganasan the maps show a continuous area including the area to the coast east of Lake Taimyr and north of Nordvik island with sources "Dolgix 1963, Popov 1948, Brykina & Gusev 2015, Wagner-Nagy 2018:3". None of these sources sanction this chunk as Nganasan settled. In particular, the extremely detailed maps by Dolgix (temporally differentiated) and Popov (summer-winter differentiated) --- based on their own extensive fieldwork --- stop just east of lake Tajmyr, not including the NE area up to the coast. You could of course say that area might have been hunting grounds and certainly no other ethnic group lay claim to it, but then the entire northern Tajmyr could just as well be mapped as Nganasan as hunting grounds (Golovnev 1999 says this explicitly). Also, following Popov, the settlements were always along drainages, and the interior areas hunting grounds, but the maps show continuous areas but not including *all* hunting grounds. What is the actual intent? 

R: As mentioned above there really are many opinions about the “right” distribution of each language and Nganasan is a good example of how much different sources vary. Nganasan speakers have traditionally relied on nomadism, which also means a mobile way of living compared to the sedentary lifestyle common in the southern regions, which also are more densely populated areas. These differences in ethnic groups’ mobility as well as in population densities have been mapped with very variable methods through history. 

In this work, the idea was first to collect different opinions to the same database and then plot these different opinions on the map so that experts can evaluate their accuracy concerning the past and present distributions with complementary knowledge they had. Based on all collected information they made their decision on how to define a particular language on the map. The sources that have been mentioned to be used as a basis for a new state-of-the-art map are listed alongside the figures. In general, if some source has been mentioned, it means that in some way it has been used for creating a new map even though it can be inconsistent with some other source in the same source list. The selected expert(s) made the decision, which are the sources that should be listed in each case. 

To indicate how different sources have been used to create state-of-the-art maps we added a short note in the manuscript (lines 314–316). When we will release a new version of S2 Appendix (Geographical database of the Uralic languages) in Zenodo we will also explain this aspect with more details. Lastly, we also added a reference to the Oxford Guide to the Uralic Languages chapter (in print) where we introduce the map-making process more in-depth (line 288). To validates its usage as a reference here and to ensure that reviewers and editor has an access to it, we provided a pre-print of this chapter in Academia.edu repository: https://www.academia.edu/70120154/Mapping_the_distribution_of_the_Uralic_languages

C: Another example is South Saami whose northen dialect on the map includes Södra Tärna with the sources "Hasselbrink 1981–1985, Rydving 2008: 360–361, Rydving 2016, Maja Lisa Kappfjell & Jussi Ylikoski (p.c.)" but Rydving is explicitly arguing for South Tärna to be counted as Ume Saami. What was the reasoning here and what informaed was asked of or provided by Ylikoski?

R: This is an important remark from the reviewer: reference to Rydving 2016 is unfortunately misleading. The South Saami map was created based on these sources, but we now realize that Rydving 2016 is misleadingly included in the list (it did contribute to our work with this map, but we do not yet fully agree with it). We will remove Rydving 2016 from the source list when releasing a new version of the database (S2 Appendix) in Zenodo. As to the subject matter itself, it may be noted that Rydving 2016 explicitly argues against the traditional view on the South Saami – Ume Saami border, which our map describes. While we (Ylikoski) sympathize with Rydving’s arguments, we consider the question still unsettled, and will try to describe the border as less definite in future releases.

From the description of ‘S2 Appendix’: “The significant impact of expert(s) have been indicated with personal communication (p.c.) label in the sources. Personal communications were used when creating state-of-the-art distributions for the languages. In these cases, the received information is often unpublished, and it is implemented to the new language distributions in different ways.”

C: The bottom line is, there are two choices: (i) Either the paper stays without a definition of the intended areas to map and/or a methodology for using the experts, but then the authors have not defined a "practice", let alone a "best practice", and the title should be changed accordingly, or (ii) such a "best practice" is defined and included in a revision. This is the only important point, everything else is essentially fine and the resource as such is excellent.

R: The intended areas and the expert evaluation process are now much more in detail explained in the manuscript (see the changes above). We also sharpened the message that “the development of the actual standardization of language area is beyond the scope of this work, and the distributions of the languages are presented exactly as they were defined in the original publications” (lines 122–124). In practice, this means that language area standardization is not part of the “Best practices workflow”. Instead, the actual content of “Best practices workflow” is comprehensively described in the final paragraph of ‘Introduction’, through the ‘Methods’, in the first part of ‘Results’, and in the ‘Discussion’. Therefore, we preserve the original title for this article. 

C: Non-important issues:

* Rephrase: "In our case, it was obvious that different opinions about the language distributions vary notably between the different sources by language."

* "but the workflow is applicable to other language families or geographical areas as well"

There are no experts for every language available for most families and areas

so this workflow is not applicable in the same form for others.

R: It is true that expert evaluations as a method to produce state-of-the-art language areas and maps is not applicable as such to all language families or geographical areas. However, we are not claiming that this methodological guideline is applicable to all other language families or geographical areas. Further, this sentence refers to the whole Best practices workflow, not only to the part where we introduced the state-of-the-art map procedure. We however added clarifications to describe the applicability of the expert evaluation process in ‘Methods’ (lines 278–279) and ‘Results’ (lines 372–377).

C: * It would be nice of more detail (or a reference) were given on how to best digitize printed maps

R: We made a few clarifications into the paragraphs where digitization was presented (lines 184–190) and added two references for a more detailed description of the process (line 181).

C: * Imperatorskoe Russkoe Geograficheskoe, Obshchestvo -> remove comma

* Imperatorskogo Russkogo Geograficheskogo Obshchestva -> why genitive?

R: Both of the sources have been corrected -> Imperatorskoe Russkoe Geograficheskoe Obshchestvo 

C: * Some Russian sources are in Roman, some in Russian majuscule and some

in Russian minuscule. Harmonize.

R: All Russian sources have been harmonized to the ‘References’. The whole reference list is also updated.

Reviewer #2:

Comment: The idea of publishing a best practice example of handling spatio-linguistic data is quite good and the relevance of this publication should not be underestimated. Good practice guidelines for data handling is available in all natural sciences and is also available in some disciplines of the humanities. A general good practice approach should be part of the digital humanities but also requires specific components for each topic. This article combines general aspects of handling geospatial data and specific aspects of linguistics. The current review focusses on the first part.

Though, the workflow employs rather general steps, it can be helpful for considering all steps and after all, too narrow standards are rather preventing the concept from being applied. In particular pointing at the importance of open licences (cc) and open platforms (zenodo) is an important consideration for up to date research. Another important point is to acknowledge certain redundancies such as offering the geospatial data as well as the ready made maps in order to allow benefits for different target groups.

Response: This comment is a good summary why different parts of the Best practices guideline are relevant in this context. The comment also helped us to understand that the ‘Data collection and harmonization’ chapter was focused on the Uralic case study instead of being a guideline in general. Therefore, we made the following changes: We updated Table 1 to cover more widely our recommendations for geospatial data production instead of concentrating only on the Uralic case study (changes in Table 1 are highlighted). We separated the columns for general level (Advisable types/features) and what types/format was used in the Uralic case study (Selected in the case study). ‘Temporal divisioning’ and ‘Metadata description and file naming’ were the new features in Table 1, and thereby included as a part of ‘Data collection and harmonization’ standard. We updated the table caption (lines 205–210). We also added a new paragraph where we explain these steps on a more general level (lines 225–240). 

In practice, this means that we now introduce, for example, a wider palette of geometry types, data formats and ways to execute temporal divisioning when harmonizing the geospatial data. This guarantees that a developed practice takes more broadly into account the differences between language families and geographical regions and makes the overall process more flexible and applicable.

The comment actually was very valuable and put us to think about different parts of practice from multiple aspects. We truly feel that it helped us to further develop the whole practice section.

C: The part on geospatial data can be further improved.

- It seems clear why polygons should be used, but why are four point data sets part of the database? It seems that this fact destroys the consistency of the database.

R: This comment was also very useful and made us think about the presentation of language distributions in general and from different aspects. It is true that polygons should be primary options to present areal information of the languages. However, there are some exceptions when the usage of points is well-reasoned, for example in cases where spatial information of languages needs to be presented at an ultimately general level (e.g. political reasons or privacy) or the occurrence of language is strongly concentrated on settlement centers such as villages. In addition, many earlier published data on languages is represented with coordinates why point geometry type should not be totally rejected. Concerning the structure of the database, we do not see that different geometry types destroy its consistency - even more we see them to complement each other. 

In summary, we made the following changes to the manuscript:

- Better justification for the use of point data (lines 159–161, Table 1 -> Changes in ‘Geometry type’).

- Decision to recommend polygons as the primary option to present language distributions, but in some exception possibility to use points alongside them (lines 232–235, Table 1 -> changes in ‘Geometry type’).

C: - I understand why WGS84 is used but for mapping and geospatial analysis projected coordinate systems are usually preferred. It would be helpful to explain the decision.

R: The decision to select the WGS84 coordinate system is based on the fact, that it is a standard in digital databases and map user interfaces in linguistics and other disciplines studying human history. The compatibility and usability were the main factors we emphasized when making this decision, but the other options were also considered in our data production process. The usage of geographic coordinate systems is justified when the study area is large (global or continental-wide), which often is a case with the language families. Projected coordinate systems could be an alternative if the data cover smaller geographical regions (e.g. an area of a small country), but in general level this seldom happens.

As a result of the comment, we better justified the selection of WGS84 in the manuscript (lines 184–187, Table 1). 

C: - The decision for shape files is not understandable at all. It is true that shape files are still wide spread but they are based on an completely outdated technology. A new god practice guideline and a new workflow should be based on up to date and sustainable technology. For small data as in this case, a text and WKT or EWKT based format such as csv would have the benefit of being easily to integrate in full reproducible research workflows. Further more, this formats are as software-independent as possible. For larger data spatialite databases and derivates such as geopackage would be a good choice. It would be helpful to discuss this point and to refer to newer developments in geography and geospatial technology. 

R: In the field of geoinformatics there is a lot of discussion about “right and wrong” data formats - some prefer one format and another something else. Further, geospatial techniques are rapidly developing and there definitely is no consensus which format is the best and which one should be avoided. However, we took the criticism concerning the usage of shapefile seriously and carefully examined the pros and cons of the different data formats. From some points of view shapefiles can be outdated technology and compared to newer formats it may have some deficiencies, for example the comment of software independence was valid. On the other hand, shapefile really is a widely used and easily convertible format, which can be used with all common GIS softwares. It also usually works well as such in the softwares developed for the geostatistical analyses, and if not, shapefiles are easy to convert to some other format.

However, when the aim is to produce the “Best practice” for the data format, which will be based on the best possible technology also in the future, the recommendation should be thought of closely. Therefore we decided to modify our practice to be more permissive. Instead of selecting only one format to be used, we decided to recommend any format which fulfills the requirements of being interoperable, up-to-date and convertible, but emphasizing widespread use and notoriety in our procedure. After careful review we decided to mention only a few exemplary formats as a recommendation including those which have a long tradition to be used in GIS and those which are based on newer technology and fit with OGC (Open Geospatial Consortium) standards: “e.g. SHP, GEOJSON and WKT” (Table 1). The same content was added to text (lines 225–232, especially 230–232). 

Finally, we wanted to emphasize the possibility to utilize the developed spatial data platform URHIA also in the file format conversion. Therefore we added a mention of the possibility to download the datasets in a variety of formats (lines 353–354).

C: - An example of overlapping data would be helpful and a more detailed explanation of how to combine data and handle the overlapping would be appreciated. 

R: An example of overlapping data is presented in Fig 1 (former Fig 2) titled as “Geographical overlap of different source materials concerning the distribution of the Khanty language(s) at the beginning of the 20th century”. This figure points out how differently separate sources of Khanty language determine the language distribution on the map. It also indicates the overlap quite illustratively. On the other hand, we have also explained the different options on how to handle overlap when creating consensus maps based on them in 1) general level (lines 248–264, more detailed explanation in Haynie & Gavin 2019 [24]), and 2) concerning our solution in Uralic case study in chapter ‘Creating state-of-the-art maps based on the digitized data and new expert opinions’. In our case, consensus information on languages was collected using an expert evaluation process where an expert or group of experts defined the distribution of a particular language based on earlier sources and the other information they collected or had. Noteworthy, the expert evaluation process is explained in more detail after the comments from reviewer 1. See also S1 Appendix, which is a new Supplementary material exemplifying the query format sent to the expert at the beginning of the process. 

C: Another possible improvement could address the handling of chronological information. Perhaps, a more general approach would make sense. Categories such as "traditional" might make sense in a specific case but are rather confusing and imprecise in general. Examples of how to handle temporal data are known from temporal data bases and from archaeology.

R: The reviewer is right, especially concerning the more general approach when determining the way to handle temporal divisions in historical geospatial datasets. We edited our manuscript concerning the standards of temporal divisioning (lines 235–238, Table 1). Now we take into consideration the variability of the different datasets: data can be temporally precise or inexact when different classifications are well-reasoned. In practice, we recommend exact dating for the data whenever possible (often present-day data), but more general divisioning when the source materials do not allow more (just as in our case study and with many other historical or prehistorical data). Even then, it would be good to strive to target century level (e.g. 19th century, 20th century or 1800, 1900 etc.) rather than more abstract or general classes in temporal structure. Then the data is more probably applicable with the data from other disciplines.

Academic Editor:

Comment: Although both reviewers judge the revisions needed to be minor they address different aspects, so the combined amount of suggested revisions would be more than minor. Both reviewers point to some cases where a particular practice might be improved or where a suggestion is not perceived as being optimal. No doubt, there is a room for different opinions about what is a best practice, so more discussion of alternative could be added. And sometimes there is still room for improvement in your own practice--it may actually strengthen the paper to admit for this possibility where relevant.

Response: We see that the comments really helped us to understand the deficiencies in the manuscript and they led to the following quite remarkable editions to the text: We explained the key concepts (e.g. language area, who was seen as an expert) more detailedly. We also added several clarifications concerning the digitalization and expert evaluation processes. Importantly, we made notable changes to ‘Data collection and harmonization’ for balancing the general aspect and Uralic case study in geospatial data creation including also the wider and more flexible recommendations for suitable file formats, geometry types and temporal divisions (see Table 1 and lines 225–240). The different aspects of the Best practices are now also better taken into account, and most comprehensively these are opened up in ‘Methods’ which from our point of view was the optimal section to deepen these aspects. However, some clarifications were also made to the ‘Results’ and ‘Discussion’.

---

## [Decision Letter · Decision Letter 1]

26 May 2022

Best practices for spatial language data harmonization, sharing and map creation – a case study of Uralic

PONE-D-21-40182R1

Dear Dr. Rantanen,

We’re pleased to inform you that your manuscript has been judged scientifically suitable for publication and will be formally accepted for publication once it meets all outstanding technical requirements.

Kind regards,

Søren Wichmann, PhD

Academic Editor

PLOS ONE

Additional Editor Comments (optional):

Reviewers' comments:

Reviewer's Responses to Questions

**Comments to the Author**

1. If the authors have adequately addressed your comments raised in a previous round of review and you feel that this manuscript is now acceptable for publication, you may indicate that here to bypass the “Comments to the Author” section, enter your conflict of interest statement in the “Confidential to Editor” section, and submit your "Accept" recommendation.

Reviewer #1: All comments have been addressed

Reviewer #2: All comments have been addressed

2. Is the manuscript technically sound, and do the data support the conclusions?

Reviewer #1: Yes

Reviewer #2: Yes

3. Has the statistical analysis been performed appropriately and rigorously? 

Reviewer #1: N/A

Reviewer #2: Yes

4. Have the authors made all data underlying the findings in their manuscript fully available?

Reviewer #1: Yes

Reviewer #2: Yes

5. Is the manuscript presented in an intelligible fashion and written in standard English?

Reviewer #1: Yes

Reviewer #2: Yes

6. Review Comments to the Author

Reviewer #1: Comments have been adequately addressed.............................................................

Reviewer #2: Many thanks for the revised version. All comments have been addressed and all issues are solved.

7. PLOS authors have the option to publish the peer review history of their article (what does this mean?). If published, this will include your full peer review and any attached files.

Reviewer #1: **Yes: **Harald Hammarström

Reviewer #2: **Yes: **Oliver Nakoinz

---

## [Editor Report · Acceptance letter]

30 May 2022

PONE-D-21-40182R1 

Best practices for spatial language data harmonization, sharing and map creation – a case study of Uralic 

Dear Dr. Rantanen:

I'm pleased to inform you that your manuscript has been deemed suitable for publication in PLOS ONE. Congratulations! Your manuscript is now with our production department. 

Kind regards, 

on behalf of

Dr. Søren Wichmann 

Academic Editor

PLOS ONE